# Lower Zinc but Higher Calcium Content in Rodent Spinal Cord Compared to Brain

**DOI:** 10.3390/cells14120922

**Published:** 2025-06-18

**Authors:** Alma I. Santos-Díaz, Brandon Bizup, Ana Karen Pantaleón-Gómez, Beatriz Osorio, Olivier Christophe Barbier, Thanos Tzounopoulos, Fanis Missirlis

**Affiliations:** 1Department of Physiology, Biophysics and Neuroscience, Cinvestav, Mexico City 07360, Mexico; alma.santos@cinvestav.mx (A.I.S.-D.); bosorio@fisio.cinvestav.mx (B.O.); 2Pittsburgh Hearing Research Center, Department of Otolaryngology, University of Pittsburgh, Pittsburgh, PA 15201, USA; bbizup@pitt.edu (B.B.); thanos@pitt.edu (T.T.); 3Department of Toxicology, Center for Research and Advanced Studies of the National Polytechnic Institute (CINVESTAV), Mexico City 07360, Mexico; ana.pantaleon@cinvestav.mx (A.K.P.-G.); obarbier@cinvestav.mx (O.C.B.)

**Keywords:** central nervous system, confocal microscopy, copper, elemental analysis, immunofluorescence, iron, metallomics, transgenic mouse, zinc transporter, zincosomes

## Abstract

Metal ion measurements using inductively coupled plasma optical emission spectroscopy revealed twofold-higher zinc content in rat brain compared to spinal cord. One hypothesis to explain this difference is the high prevalence of synapses that corelease glutamate and zinc in the brain, marked by the vesicular Zinc Transporter-3 (ZnT3). In contrast, spinal cord tissue showed significantly higher calcium content, reflecting calcifications in the arachnoid. The above observations were made in 60-day-old adult male and female rats fed ad libitum or a restricted diet. In this study, we asked if the calcium and zinc content of the brain and spinal cord was species-specific or evolutionarily conserved, and whether the distinct concentration of zinc in the brain and spinal cord resulted from a different expression pattern of ZnT3, the primary transporter in synaptic vesicles. To address these questions, we examined 8-week-old wild-type male and female mice raised under conventional laboratory conditions and used a knock-in mouse that expresses a human influenza hemagglutinin epitope tag at the C terminus of the endogenous *ZnT3* gene to assess the transporter’s abundance in spinal cord sections. Our results show conserved inverse differences in zinc and calcium content in mouse brain and spinal cord, but detectable ZnT3 signal in spinal cord. Whereas vesicular zinc modulates glutamatergic and GABAergic signaling and sensory processing, the functional significance of calcium aggregates in the arachnoid remains unknown.

## 1. Introduction

The distribution of zinc [1], copper [2], and iron [3] in brain has attracted attention due to the involvement of these metal ions in neurodegenerative disorders [4,5,6]. X-ray fluorescence imaging and other elemental analysis techniques permit the quantification of these three and other elements in parallel, within brain tissue [7,8,9]. The spinal cord is an extension of the central nervous system which has a key function in coordinating motion, loss of which is evident in motor neuron disease or following mechanical injury [10,11,12]. Despite recognition that metal ion homeostasis is relevant in the pathophysiology of neurodegeneration [13,14], few studies have analyzed metal ion concentrations in the spinal cord [15,16,17]. It is therefore not known whether metal ions accumulate to a similar extent, and with similar cellular and subcellular localizations, in the spinal cord and brain.

Our own research in this area began with measurements of rat tissue [18]. The first result that immediately stood out was a significant accumulation of calcium in the spinal cord. A literature review revealed the presence of calcifications in the arachnoid; this was first identified by Weed in 1920 [19] and described in detail by Herren in 1936 [20]. The arachnoid is a thin cellular layer that covers the brain and spinal cord [21]. Whorls of arachnoid cells participate in the formation of the calcified deposits [20]. No function was attributed to these structures, which were noticed by various researchers studying spinal cord disease [22,23,24,25,26] and even classified according to their size [27]. Notably, the early work identified these structures in healthy subjects [20]. We also confirmed their presence with X-ray fluorescence imaging in spinal cord sections from untreated adult male rats [18].

The second notable finding which we obtained from our multi-organ metallome analysis was significantly lower content of zinc (assessed per dry weight) in spinal cord compared to brain [18]. There is a long history of studying zinc in the brain [1]. Already from the 1960s, Haug extended Timm’s sulfide silver procedure for use in electron microscopy and visualized with fine detail zinc accumulations in synaptic boutons of hippocampal mossy fibers [28]. The preferential accumulation of zinc in the hippocampus was verified with radioactive zinc detection [29], spectroscopy, and atomic absorption [30]. Since then, synaptically released zinc has emerged as a major neuromodulator that shapes neurotransmission and sensory processing in many brain regions, including the hippocampus, cortex, and brainstem [1]. The maintenance of synaptic zinc is dependent on the vesicular zinc transporter ZnT3 [31]. A ZnT3 knock-out mouse showed lack of synaptic zinc accumulation in the mossy fibers and 20% less zinc in the hippocampus and cortex where the transporter is predominantly expressed [32]. Similar observations were made in brains of mice lacking the cellular machinery that delivers ZnT3 to the synaptic vesicles [33], which is evolutionarily conserved in invertebrates [34]. The functional significance of synaptic zinc has been evaluated in numerous studies using ZnT3 knock-out mice, leading to the proposal that zinc acts on different receptors as both a pre- and post-synaptic neuromodulator to regulate neurotransmission, synaptic plasticity, and sensory processing [1,35,36,37,38,39,40,41,42,43,44,45,46,47,48,49,50,51,52,53].

We developed a method to correlate spatially resolved quantitative measures of zinc by X-ray fluorescence imaging (a technique previously used by others to confirm the role of ZnT3 in zinc accumulation within the mossy fibers [54]) to whole-tissue inductively coupled plasma optic emission spectroscopy (ICP-OES) data [18]. The surprising outcome of this analysis was that over a third of all synapses in rat brain were predicted to be of the glutamate–zinc type [18]. Our finding coincided with another unbiased observation based on proteomic and imaging analysis of single synaptic vesicles isolated from rat brain, which concluded that the most abundant (34%) synaptic vesicle type co-expressed vesicular glutamate transporter 1 (VGLUT1) and ZnT3, whereas 25% were of the simple glutamatergic (VGLUT1-positive, ZnT3-negative) type [55]. Based on these findings, we proposed that VGLUT1-ZnT3 synaptic vesicles might be more abundant in rat brain compared to rat spinal cord [18].

One of the limitations of our prior work was that it was performed exclusively in a single laboratory animal species, the rat. We therefore readdressed these findings in mice to see if the relative changes in zinc and calcium content in the two parts of the central nervous system were evolutionarily conserved in the two rodent species. We carried out ICP-OES analysis of brain and spinal cord in mice of both sexes and compared the new data to our prior findings for rat. To address whether lower zinc in the spinal cord was indeed due to the absence of VGLUT1-ZnT3 synaptic vesicles in this tissue, we used a newly developed ZnT3-HA knock-in mouse with a hemagglutinin epitope tag attached to the C terminus of the endogenous ZnT3 gene to assess the transporter’s abundance in spinal cord sections [56]. Our results, however, suggested that ZnT3 is also expressed in the spinal cord. We therefore considered the alternative possibility that the lower zinc content per g dry weight in spinal cord versus brain reflects a higher proportion of axonal tracks to cell bodies in the former tissue.

## 2. Materials and Methods

### 2.1. Animals and Sample Preparation

The Institutional Committee for the Care and Use of Laboratory Animals (Comité Interno para el Cuidado y Uso de los Animales de Laboratorio, CICUAL) from Cinvestav approved all animal procedures (protocol number: 0188-16). All experimental procedures were conducted according to the current Mexican legislation NOM-062-ZOO-1999 (SAGARPA) and in agreement with the Guide for the Care and Use of Laboratory Animals of the National Institutes of Health (NIH).

Eight-week-old mice of the c57bl/6 strain were used. These animals were control subjects, part of an unrelated parallel experimental protocol that excluded any assessment of the central nervous system, and they were therefore used here to optimize animal use in biomedical research. We only used untreated animals or animals that had received a single subcutaneous injection of 10 µL phosphate buffered saline (PBS) 4 h or 24 h prior to their sacrifice. The latter group of animals was included to increase the number of individual animals assessed and because there was no *a priori* reason to expect changes in the metallome following injection of PBS. The animals were anesthetized with isoflurane (Sofloran Vet, PISA Farmaceutica, Hidalgo, Mexico) using V-1 Table Top Lab Animal System and euthanized by exsanguination. The animals were placed on ice while brain and spinal cord were extracted. The organs were sectioned and stored at −80 °C. Organ pieces were freeze-dried for 36 h and then pulverized with liquid nitrogen.

ZnT3-HA mice were used for ZnT3 protein localization. ZnT3-HA mice were generated on a C57Bl/6 background using CRISPR/CAS9 to insert a human influenza hemagglutinin (HA) tag in frame at the carboxy terminus, as previously described [56]. Control mice for these experiments were wild-type littermates of ZnT3-HA mice. Eight-week-old mice were deeply anesthetized using isofluorane and transcardially perfused with ice-cold phosphate-buffered saline (PBS) and then with 4% paraformaldehyde (PFA) in PBS. The brains and spinal cords of the mice were then extracted and post-fixed in 4% PFA for 2 h before being washed in PBS and then cryoprotected in a series of 15% sucrose and 30% sucrose for 24 h each. Tissues were then embedded in Optimum Cutting Temperature (O.C.T.) compound (Scigen Tissue-Plus, Fisher Scientific, Waltham, MA, USA, Cat3 23-730-571) at −20 °C before cryosectioning. Serial cryosections from ZnT3-HA brains and spinal cords were taken at a thickness of 35 µm and collected into PBS + 0.01% sodium azide until staining.

### 2.2. Inductively Coupled Plasma Optic Emission Spectroscopy

A 20 mg amount of dry-weight tissue powder was digested by adding 1 mL of concentrated, metal-free nitric acid and using the animal tissue program of the MARS 6 CEM microwave, which gradually increases the temperature of the sealed containers to 200 °C and maintains it stable for 15 min, followed by a cooling phase. Clear digested samples were then calibrated to 5 mL with Milli-Q water. Elemental analysis was performed by ICP-OES on a PerkinElmer Optima 8300 instrument. Levels of total Ca, Mg, P, Zn, Fe, and Cu content were determined by extrapolation against calibration curves of standard solutions of known concentration. The sample digestion process and elemental determinations were carried out in triplicate for each organ, with the average value of these measurements being used as a high-confidence single data point per organ per mouse for subsequent reporting and analysis.

### 2.3. Statistical Analysis of the ICP-OES Results

The ICP-OES data were classified by element, organ, sex, and whether animals had or had not received a mock (PBS) injection. In this way we generated three groups (untreated male control animals, untreated female control animals, and male mock-injected control animals). As the main focus of comparison in this study was whether there are differences in elemental content between brain and spinal cord, we used Student’s t-test in two different ways. First, we asked whether there was a statistically significant difference when comparing the mean values for each element in the brains of the above-mentioned groups to those in the respective spinal cords (Table 1; a similar approach was used to reanalyze data from [18] shown in Table 2 for a direct comparison between the two studies). Second, we pooled all the individual measurements per organ per rat and applied Student’s *t*-test, albeit with a significantly higher number of data points per condition (Figure 1). These statistical tests were performed with GraphPad Prism software version 5.0.

### 2.4. ZnT3-HA Imaging by Confocal Microscopy in Brain and Spinal Cord Sections

Cryosections were washed 3 × 10 min in 1X PBS containing 0.3% Triton X-100 (0.3% PBST), blocked for 1 h at room temperature in 0.3% PBST with 5% normal goat serum, and then incubated in primary antibodies (Rabbit anti-HA tag, 1:1000, Cell Signaling Technologies Cat# 3724S) diluted in blocking buffer overnight at 4 °C. Cryo-sections were then washed 3 × 10 min in 1X PBS and incubated in the secondary antibody solution diluted in 1X PBS (Goat anti-rabbit AlexaFluor 555, 1:000, Invitrogen Cat# A21429; DAPI, 1:5000; Thermo Fisher Scientific Cat# D1306). Sections were washed 3 × 10 m in 1X PBS, mounted onto slides, and coverslipped with Prolong Gold Antifade Mountant (Thermo Fisher Scientific, Cat#P36930). Fluorescent images were captured using a Leica Stellaris confocal fluorescent microscope using a 10× air objective. Laser power and gain settings were held constant for each experiment, and images were stitched with 10% overlap using Leica LASX software (version 5.0.2).

## 3. Results

ICP-OES was performed to obtain an elemental analysis of brain and spinal-cord dry tissue from untreated 8-week-old mice or mice injected with a phosphate buffered solution. These animals were taken from control groups of an experiment that will be reported in another paper; for the present study, they were used to assess biologically relevant transition and alkaline earth metal ions previously shown to differ in their concentrations between the two main parts of the central nervous system. Table 1 shows the results from this analysis.

Results were broadly similar for all elements in untreated male and female animals and in the mock-injected group. A seven-fold increase in calcium content was present in the spinal cord compared to the brain (*p* = 0.03). Zinc and copper were 40% less abundant in spinal cord compared to brain (*p* = 0.01 and *p* = 0.004, respectively). At this level of analysis, no statistically significant differences between the two organs were observed in terms of magnesium, phosphorus, or iron content.

All data obtained for a brain or spinal cord of individual mice, irrespective of sex or treatment, were plotted together (Figure 1). In this graph, the mean value for brain calcium was 0.3 ± 0.2 mg per g dry weight compared to 2.9 ± 2.0 mg per g dry weight in the spinal cord, a highly significant result (*p* = 2 × 10^−9^) despite notable variability between animals. This difference represented an eight-fold enhancement for this element in spinal cord compared to brain, confirming previous observations in rat [18]. Magnesium content was determined to be 0.63 ± 0.08 and 0.54 ± 0.17 mg per g dry weight in brain and spinal cord, respectively, indicating a 14% difference. Phosphorus at 9.0 ± 2.8 mg:g brain versus 11.3 ± 3.8 mg per g spinal cord showed a 25% higher value in the spinal cord, again consistent with previous observations in rat [18]. In contrast, but as expected in light of the previous work, zinc content was 41% lower in the spinal cord (0.025 ± 0.009 mg:g dry weight) compared to the brain (0.042 ± 0.014 mg: g dry weight). Surprisingly, however, both iron (0.033 ± 0.012 *versus* 0.044 ± 0.016) and copper (0.009 ± 0.004 versus 0.015 ± 0.004) were found to be 25% and 41% lower, respectively, in the spinal cord compared to the brain (Figure 1).

The findings that calcium content was seven-to-eight-fold higher and phosphorus content roughly 25% higher in spinal cord versus brain, while zinc content was roughly 40% lower, corroborated the situation in the rat central nervous system [18]. Our observations of roughly 40% less copper, 25% less iron, and 14% less magnesium led us to reanalyze the data from our earlier study, which were conveniently obtained in four experimental groups, dietary-restricted versus untreated fed ad libitum male and female 60-day-old rats (Table 2).

**Table 1 cells-14-00922-t001:** Comparison of metal content (mg element per g tissue dry weight) between murine brain and spinal cord (n refers to the number of animals analyzed per group).

Experiment	Organ	Ca	Mg	P	Zn	Fe	Cu
#1 male untreated (n = 11)	Brain	0.4 ± 0.2	0.64 ± 0.08	9.4 ± 3.1	0.041 ± 0.015	0.047 ± 0.014	0.016 ± 0.005
	Spinal cord	4.2 ± 2.3	0.68 ± 0.18	13.7 ± 3.2	0.030 ± 0.010	0.040 ± 0.013	0.008 ± 0.003
#2 male mock inj.* (n = 11)	Brain	0.2 ± 0.1	0.67 ± 0.08	10.1 ± 2.9	0.040 ± 0.017	0.046 ± 0.010	0.013 ± 0.004
	Spinal cord	2.4 ± 1.4	0.48 ± 0.14	11.5 ± 3.9	0.020 ± 0.007	0.033 ± 0.011	0.010 ± 0.005
#3 female untreated (n = 8)	Brain	0.4 ± 0.3	0.57 ± 0.03	7.0 ± 0.3	0.047 ± 0.005	0.039 ± 0.023	0.015 ± 0.001
	Spinal cord	1.8 ± 1.2	0.44 ± 0.04	7.7 ± 0.7	0.025 ± 0.003	0.024 ± 0.005	0.007 ± 0.001
Mean (n = 3)	Brain	0.3 ± 0.1	0.63 ± 0.05	8.8 ± 1.7	0.043 ± 0.004	0.044 ± 0.004	0.015 ± 0.001
	Spinal cord	2.8 ± 1.1	0.53 ± 0.13	11.0 ± 3.1	0.025 ± 0.005	0.032 ± 0.008	0.009 ± 0.001
Fold difference		7.1	−0.2	0.2	−0.4	−0.3	−0.4
T-test (*p*-value)		0.03	0.30	0.35	0.01	0.09	0.004

* This group received a subcutaneous injection of phosphate saline solution 4 h or 24 h prior to sacrifice.

**Table 2 cells-14-00922-t002:** Comparison of metal content (mg element per g tissue dry weight) between brain and spinal cord of the rat; all data were obtained and re-analyzed from [18].

Experiment	Organ	Ca	Mg	P	Zn	Fe	Cu
#1 male ND (n = 11)	Brain	0.3 ± 0.2	0.75 ± 0.11	14.7 ± 3.4	0.063 ± 0.015	0.086 ± 0.017	0.011 ± 0.003
	Spinal cord	4.1 ± 2.0	0.73 ± 0.08	20.2 ± 4.1	0.035 ± 0.011	0.073 ± 0.037	0.009 ± 0.008
#2 male RD (n = 11)	Brain	0.3 ± 0.2	0.80 ± 0.07	16.2 ± 1.7	0.067 ± 0.015	0.100 ± 0.046	0.011 ± 0.002
	Spinal cord	2.5 ± 2.2	0.63 ± 0.08	17.8 ± 4.1	0.033 ± 0.005	0.057 ± 0.022	0.006 ± 0.003
#3 female ND (n = 10)	Brain	0.6 ± 0.6	0.73 ± 0.08	15.0 ± 1.8	0.065 ± 0.010	0.078 ± 0.014	0.011 ± 0.002
	Spinal cord	4.3 ± 3.9	0.58 ± 0.08	16.9 ± 3.0	0.028 ± 0.004	0.049 ± 0.014	0.004 ± 0.002
#4 female RD (n = 9)	Brain	0.4 ± 0.2	0.79 ± 0.06	15.6 ± 0.9	0.071 ± 0.006	0.077 ± 0.012	0.012 ± 0.002
	Spinal cord	2.8 ± 4.3	0.57 ± 0.07	15.7 ± 2.9	0.035 ± 0.016	0.051 ± 0.016	0.004 ± 0.002
Mean (n = 4)	Brain	0.4 ± 0.1	0.77 ± 0.04	15.4 ± 0.6	0.066 ± 0.003	0.085 ± 0.011	0.012 ± 0.001
	Spinal cord	3.4 ± 0.9	0.63 ± 0.08	17.7 ± 1.9	0.033 ± 0.000	0.057 ± 0.011	0.006 ± 0.002
Fold difference		7.8	−0.2	0.2	−0.5	−0.3	−0.5
T-test (*p*-value)		0.0006	0.02	0.06	0.000006	0.01	0.003

ND—normal diet (ad libitum); RD—restricted diet.

In the rat, an eight-fold increase in calcium content was recorded in the spinal cord compared to the brain (*p* = 0.0006), whereas zinc and copper were 50% less abundant (*p* = 0.000006 and *p* = 0.003, respectively). Content levels of iron and magnesium were also 30% (*p* = 0.01) and 20% (*p* = 0.02) lower, respectively, while that of phosphorus was 20% higher (*p* = 0.06, i.e., not reaching statistical significance in this analysis). The results comparing the two species are shown in Figure 2.

**Figure 1 cells-14-00922-f001:**
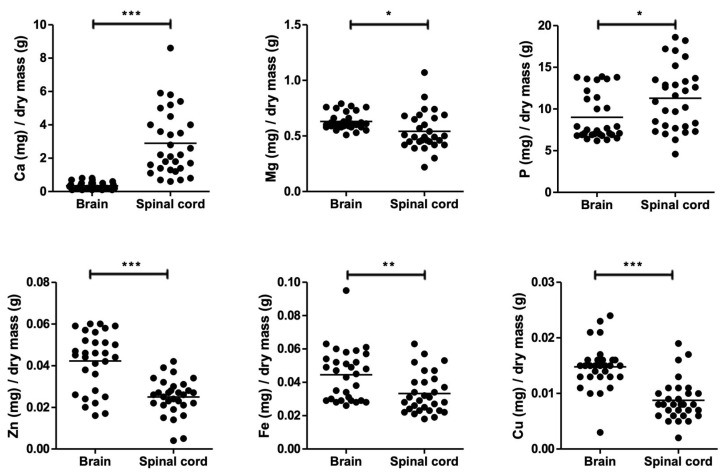
Elemental analysis of dry brain and spinal-cord tissue from 19 male mice, sacrificed 8 weeks after birth. Statistical analysis was carried out by paired Student’s T-test (*** depict *p* < 0.001; ** *p* < 0.01; * *p* < 0.05). Calcium and phosphorus content increased in spinal cord compared to brain. Conversely, content levels of zinc, copper, iron, and magnesium were lower.

**Figure 2 cells-14-00922-f002:**
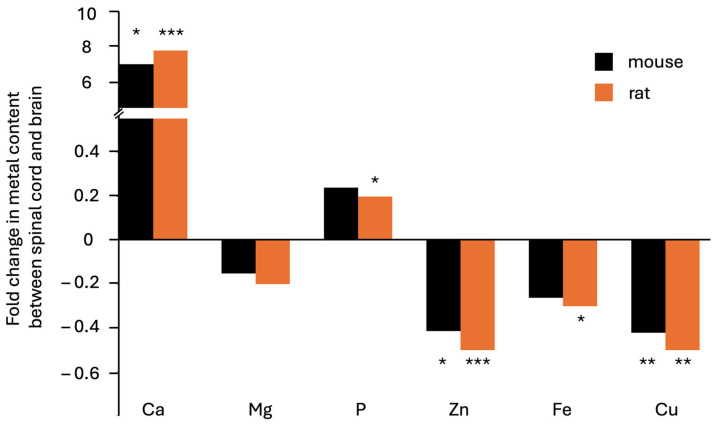
Fold differences in elemental content between spinal cord and brain in laboratory specimens from two rodent species. Data on the mouse are from this study, while data on the rat are from [18], showing the statistical analysis presented in Table 1 and Table 2 (*** depict *p* < 0.001; ** *p* < 0.01; * *p* < 0.05). By far the largest difference is an almost eightfold excess of calcium in the spinal cord in both species. All three transition metal ions shown were found in lesser proportions (decreases of between 20 and 50%) in spinal cord compared to brain.

We concluded that the spinal cord and brain showed very similar patterns in both rodent species, but there were significant differences between tissues from each species in terms of accumulation of metal ions. We discuss the significance of these findings below, after we first turn our attention to the question of whether the diminished zinc content in spinal cord compared to brain might be due to reduced or absent ZnT3 expression in this organ, as previously hypothesized [18]. To address this question, we used transgenic mice that express a human influenza hemagglutinin epitope tag at the C terminus of the endogenous *ZnT3* gene [56].

In cryosections from the brains of ZnT3-HA mice, we found that the ZnT3-HA expression pattern was consistent with previously established data on ZnT3 expression and vesicular zinc concentration and distribution, with high concentrations of ZnT3-HA in the hippocampus, amygdala, and cortex (Figure 3a, left) [38,57,58]. Conversely, brain cryosections from ZnT3-WT mice, which lack the HA epitope tag, showed no immunoreactivity, confirming the specificity of the fluorescent signal (Figure 3a, right). To determine whether ZnT3 is expressed in the spinal cord, we stained spinal cord cryosections from ZnT3-HA mice and ZnT3-WT mice. We observed clear ZnT3-HA signal in the grey matter area of the spinal cord, suggesting that the reduced zinc levels observed in the spinal cord relative to the brain are not due to an absence of ZnT3 expression, but may be due to reduced expression levels in the spinal cord relative to the brain (Figure 3b).

**Figure 3 cells-14-00922-f003:**
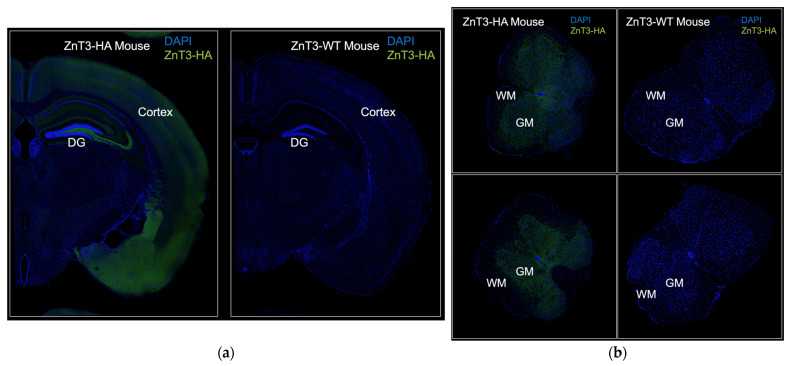
Immunofluorescent detection of ZnT3 expression in brain and spinal cord using ZnT3-HA mice. Wild-type animals (ZnT3-WT) were used as negative controls to show antibody specificity (left side of each panel). (**a**) Coronal section through one brain hemisphere. ZnT3 localizes in the dentate gyrus (DG) of the hippocampus and in the cortex. (**b**) Coronal sections through the spinal cord. ZnT3 staining is evident in the grey matter (GM) area where most neuronal cell bodies reside and absent in the white matter (WM) area. Imaging conditions were identical for panels (**a**,**b**).

## 4. Discussion

ICP-OES analysis of mouse spinal cords showed high variability in calcium accumulation in this tissue and an average of seven times more calcium per g dry mass compared to brain. Our interpretation of this finding is the working hypothesis that the spinal cord arachnoid accumulates calcium in form of aggregates in otherwise healthy individual mice, just as we previously demonstrated by X-ray fluorescence imaging in rats [18]. We anticipate that the present work will raise awareness of their existence. Despite a long history of observing these structures, formed by groups of cells belonging to the arachnoid matter [19,20,21,22,23,24,25,26,27], no physiological function has yet been attributed to such prevalent structures along the spinal cord. The high variability among calcium measurements in different mice or rat samples likely reflects high variability in the prevalence of the calcified structures and a high degree of tolerance of calcium storage in the arachnoid. It will be interesting to test whether an enhanced or diminished prevalence of calcifications correlates with other biological phenomena and determine how and when these structures develop. X-ray fluorescence imaging identified notable accumulation of phosphorus in the calcifications [18]. Thus, one might assume that the 25% increase in phosphorus in the spinal cord was also due to the presence of these structures.

In contrast, our other working hypothesis that ZnT3 would be absent from the spinal cord was refuted in this study. Indeed, the expression pattern of ZnT3 in the brain cortex and hippocampal area observed in this study closely matches the zinc accumulation in the same brain regions observed in many previous studies [18,38,57,58]. We now show the same to be true for the transporter’s pattern in the grey matter of the spinal cord, where neuronal cell bodies are mostly found, which also corresponds to the location zinc was shown to accumulate in our previous study [18]. The hypothesis that zinc content is lower in the spinal cord because of reduced presence of zinc–glutamatergic synapses could still be true and should be tested in future experiments by analyzing metallomes of brain and spinal cord from ZnT3 knock-out animals. A reduction in total brain zinc content has been shown in ZnT3 knock-out animals [32,54], but whether spinal cord zinc content is similarly affected in these animals remains untested. The proposed measurement is attractive for the additional reason that a ZnT3 homolog in *Drosophila melanogaster* has been associated with the formation of kidney stones, with zinc having been identified in the cores of such aberrant calcifications from human patients [59]. High or low zinc content has also been associated with calcifications in bone and vascular tissue [60,61]. It would therefore be very interesting to investigate a possible role of zinc in arachnoid tissue calcifications.

Independently of whether low zinc and high calcium content in the spinal cord are interdependent phenomena, the present study sounds a cautionary note for this idea because diminished zinc content in the spinal cord was found to be not specific to this metal ion. Copper was also found to be lacking to a similar degree as zinc in the spinal cord compared to the brain. To a lesser extent a similar trend of reduced accumulation in the spinal cord versus brain was observed for iron and magnesium. Thus, an alternative explanation of our findings might be limited to the differences in the relative proportions of white and grey matter in spinal cord and brain, because the spinal cord is composed of a much higher proportion of white matter (predominantly axonal tracts) compared to the brain, and X-ray fluorescence imaging suggests that metal ion concentrations are significantly lower in white matter [18]. If this simpler hypothesis for the differences observed is true, the main result that stands out from the metallome determination of the two tissues is the persistent presence of calcium in the spinal cord, which is likely attributable to the arachnoid calcifications.

To conclude, the present study offers a valuable reference for future studies measuring metal ions in the spinal cord by demonstrating important differences in the accumulation of transition and alkali earth metal ions in this tissue compared to the brain in two rodent species.

## Data Availability

All data presented in this article have been provided as a Appendix A File.

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
