# Peer review of "Lower Zinc but Higher Calcium Content in Rodent Spinal Cord Compared to Brain"

_cells, 2025, doi:10.3390/cells14120922_

Round 1

Reviewer 1 Report

Comments and Suggestions for Authors

The manuscript entitled “Lower zinc but higher calcium in rodent spinal cord compared to brain“ by Alma I. Santos-Díaz et al. presets the comparison of the metallome in mouse spinal cord compared to brain. While there is interesting data the description of the experiments is sometimes confusing and discussion is very speculative. While no additional experiments would be required a significant re-writing has to be performed. My specific points of concern are:

  1. The object of the study

While it is clear that the authors want to compare the metallomes of brain and spinal cord the animal the comparison between animal treatment is confusing. It is not explained what was the purpose of PBS injection? It is also not discussed that injection led to decrease in the content of Ca/Mg and Zn. Also it is stated that injection happened 4 or 24h prior sacrifice. Was there difference between injection time? (See also point minor corrections). Was the mean in Table 1 calculated including this results, which clearly is different than untreated animals?

Then we move to Table 2. While authors state that the data were presented in their previous study this declaration should be placed in the caption of the Table, not footnote. The table could also be minimized to data relevant to this study. The inclusion of dietary restriction is not relevant as no such treatment was included in the present study. Especially since it is not discussed why for calcium only there is a significant difference of calcium amount in the spinal cord.

  1. Molecular basis

In order to explain the difference between spinal cord and brain in terms of zinc content the authors studied the presence of ZnT3. While it is an interesting hypothesis it is very far fetched considering the presented data. It would be most suitable to make a comparison of the tissue slices between immunofluorescence and X-ray fluorescence to correlate the areas. If authors are unable to perform as synchrotron is not quickly accessible I would settle for side by side comparison with the images from the authors previous publication (Figure 5 in “Metal ion content of internal organs in the calorically restricted Wistar rat”). This is suggested in the discussion by the authors but it would be beneficial to see the results side by side. Of course and appropriate statement should be added to reveal that this data was obtained previously. I would also suggest to make an analysis of the fluorescent microscopy images. Statements like “Although ZnT3 expression is somewhat weaker based on immunofluorescence detection….” are not scientific enough considering our ability to quantify the image signal. Also the discussion can be enhanced by citing relevant data on the level o mRNA for ZnT3. Though in humans, the human protein atlas shows almost 7 fold difference of ZnT3 mRNA between the  cerebral cortex and spinal cord, which is aligned with the data presented here for mice. Finally, authors state that the difference between the zinc levels could be traced to difference in the proportion of grey vs white matter. Though this can be traced to discussion of X-ray images I would still use more papers/databses to describe the variation between expression of zinc binding proteins between the two tissues.

Minor corrections

  1. I would place Figure 1 before Table one and discuss them together.
  2. Figure 2 Y axis caption – should there be spinal cord instead of medulla?
  3. Figure 3 please add numbers or arrows to show on the image where are the parts discussed in the caption (e.g. dentate gyrus etc.)
  4. While authors state that individual data points are given in supplementary, no such file was added. Instead we have unprocessed blots, which are not stated in the manuscript (though they should be also added as supplement).
  5. The title is a bit misleading as copper and iron were also found in relevant difference between the two tissues.

Author Response

We thank the reviewer for the carefully considered comments.

Comment 1: The manuscript entitled “Lower zinc but higher calcium in rodent spinal cord compared to brain“ by Alma I. Santos-Díaz et al. presets the comparison of the metallome in mouse spinal cord compared to brain. While there is interesting data the description of the experiments is sometimes confusing and discussion is very speculative. While no additional experiments would be required a significant re-writing has to be performed. My specific points of concern are:

  1. The object of the study

While it is clear that the authors want to compare the metallomes of brain and spinal cord the animal the comparison between animal treatment is confusing. It is not explained what was the purpose of PBS injection? It is also not discussed that injection led to decrease in the content of Ca/Mg and Zn. Also it is stated that injection happened 4 or 24h prior sacrifice. Was there difference between injection time? (See also point minor corrections). Was the mean in Table 1 calculated including this results, which clearly is different than untreated animals?

Response 1: To be clearer, we have added in the Methods section a sentence and the section now reads as follows:

“Eight-weeks-old mice of the c57bl/6 strain were used. These animals were control subjects, part of an unrelated parallel experimental protocol that excluded any assessment of the central nervous system, and they were therefore used here to optimize animal use in biomedical research. We only used untreated animals or animals that received a single subcutaneous injection of 10µL phosphate buffered saline (PBS) 4 hours or 24 hours prior to their sacrifice. The latter group of animals was included to increase the number of individual animals assessed and because there was no a priori reason to expect changes in the metallome following injection of PBS.”

It is important to note that we used tissues from animals that would otherwise be disposed, as stated upfront in the Methods section cited above. The purpose of the PBS injection is therefore not relevant to this study; these are mock injections to test injections of an active substance in the animal model for a different story.

We fully understand the concern that there is an observed difference in spinal cord Mg2+ and Zn2+ between the control and PBS-injected groups; it is, however, precisely for readers to be able to observe this difference that the data are presented in two different ways. Please note that in this study we focus on the comparison between brain and spinal cord and the conclusion that Zn2+ is lower in spinal cord compared to brain is not affected by the fact that this effect appears slightly more pronounced in the PBS mock injection group.

Thus, we were faced with two choices, either to show much less information leading to a similar conclusion or to include various experimental groups and provide more than one type of statistical analysis; we believe the second approach is scientifically more robust. We hope/argue that our reporting is fully transparent and allows our study’s primary conclusion can be evaluated in multiple ways and always stands.

Last, we did not separate the animals at the 4h and 24h timepoints (after injection), because we see no changes according to time point, while the total n per group is low in the 24h group (n=7 and n=4, respectively). The mean in Table 1 was indeed calculated including the PBS-injected group as stated in the text.

Comment 2: Then we move to Table 2. While authors state that the data were presented in their previous study this declaration should be placed in the caption of the Table, not footnote. The table could also be minimized to data relevant to this study. The inclusion of dietary restriction is not relevant as no such treatment was included in the present study. Especially since it is not discussed why for calcium only there is a significant difference of calcium amount in the spinal cord.

Response 2: We have moved the statement that data shown on Table 2 were collected and presented in a previous study on its caption. The rationale for including the dietary restriction data is parallel to the rational of showing the PBS-injection data; it allows for the claim that the phenomenology we describe is generalizable beyond the specific condition when rats were grown with ad libitumaccess to chow under laboratory conditions, permitting more vigorous statistical analyses and justifying the summary of the results from both studies as presented in Figure 2.

Comment 3: Molecular basis

In order to explain the difference between spinal cord and brain in terms of zinc content the authors studied the presence of ZnT3. While it is an interesting hypothesis it is very far fetched considering the presented data. It would be most suitable to make a comparison of the tissue slices between immunofluorescence and X-ray fluorescence to correlate the areas. If authors are unable to perform as synchrotron is not quickly accessible I would settle for side by side comparison with the images from the authors previous publication (Figure 5 in “Metal ion content of internal organs in the calorically restricted Wistar rat”). This is suggested in the discussion by the authors but it would be beneficial to see the results side by side. Of course and appropriate statement should be added to reveal that this data was obtained previously. I would also suggest to make an analysis of the fluorescent microscopy images. Statements like “Although ZnT3 expression is somewhat weaker based on immunofluorescence detection….” are not scientific enough considering our ability to quantify the image signal. Also the discussion can be enhanced by citing relevant data on the level o mRNA for ZnT3. Though in humans, the human protein atlas shows almost 7 fold difference of ZnT3 mRNA between the  cerebral cortex and spinal cord, which is aligned with the data presented here for mice. Finally, authors state that the difference between the zinc levels could be traced to difference in the proportion of grey vs white matter. Though this can be traced to discussion of X-ray images I would still use more papers/databses to describe the variation between expression of zinc binding proteins between the two tissues.

Response 3:  We agree with the reviewer that our quantitative statements are not appropriate. These experiments were performed with the intent to test whether ZnT3 is expressed in the spinal cord, but not to make any quantitative claims regarding ZnT3 expression levels. As such, we have removed the statement that “ZnT3 expression is somewhat weaker based on immunofluorescence detection” from the Discussion section (Previously lines 257-258). It now states: “Indeed, ZnT3-HA is present throughout the spinal cord, albeit in the grey matter area, where neuronal cell bodies are mostly found.”

We felt, however, that bringing in Synchrotron images published elsewhere was too much borrowing from the prior study. To address this point, we have included the labeling of Figure 3 (see below) and we have also added the following explicit text to the Discussion:

“Indeed, the expression pattern of ZnT3 in the brain cortex and hippocampal area observed in this study closely matches with zinc accumulation in the same brain regions as observed in many previous studies [18, 38, 57, 58], while we now show the same to be true for the transporter’s pattern in the grey matter of the spinal cord, where neuronal cell bodies are mostly found, which also corresponds to the location zinc was shown to accumulate in our previous study [18].”

Comment 4: Minor corrections, I would place Figure 1 before Table one and discuss them together.

Response 4: This point was addressed above; some readers will prefer the simplicity of Figure 1 (but where each data point corresponds to a metal determination from a different animal independently of sex or treatment) whereas other readers may wish to pay attention on subtle differences by sex or PBS injection, which were pointed out by this careful reviewer.

Comment 5: Figure 2 Y axis caption – should there be spinal cord instead of medulla?

Response 5: Thank you for noticing, we have corrected medulla to spinal cord.

Comment 6: Figure 3 please add numbers or arrows to show on the image where are the parts discussed in the caption (e.g. dentate gyrus etc.)

Response 6: Thank you for this suggestion, we have revised Figure 3 and its legend to include the indications of dentate gyrus etc.

Comment 7: While authors state that individual data points are given in supplementary, no such file was added. Instead we have unprocessed blots, which are not stated in the manuscript (though they should be also added as supplement).

Response 7: We are unsure what happened here. The uploading system did not permit an excel file and so it was converted into a pdf file, but still all individual data points were given in that file.

Comment 8: The title is a bit misleading as copper and iron were also found in relevant difference between the two tissues.

Response 8: We have chosen to address this comment with the graphical abstract, which indicates “zinc and other transitional metals” are lower in the spinal cord. We have kept the title for three reasons: 1) the study was performed to test whether the finding observed in rats is also present in mice (which it is); 2) further, the study investigates whether ZnT3 is responsible for the difference reported (which it is not) and hence there is significant emphasis on zinc; 3) the manuscript was therefore submitted to a special issue on zinc in neuroscience.

Reviewer 2 Report

Comments and Suggestions for Authors

The authors were sought to investigate the metals distribution in adult mouse brain and spinal cord. Given that the same group has conducted more extensive observation in rat (Ref 18), this mouse study adds little insight in this filed. Here are some comments for improving the manuscript.

  1. Introduction needs to be reorganized. Current version did not deliver sufficient rationale for repeating the same research from rat to mice.
  2. Provide ref of “In cryosections from the brains of ZnT3-HA mice, we found that the ZnT3-HA expression pattern was consistent with previously established data on ZnT3 expression and vesicular zinc concentration and distribution, with high concentrations of ZnT3-HA in the hippocampus, amygdala, and cortex (Figure 3a, left).”
  3. The authors claim that: “The working hypothesis that the spinal cord arachnoid accumulates calcium aggregates in …is supported by the ICP-OES analysis presented in this study.” However, I didn’t find this evidence in the present study. The arachnoid was not isolated.
  4. Fig 2. Medulla or spinal cord? This figure looks confused in the current version, needs to be represented.
  5. Fig 3. The cerebellum is a major repository of metals, how about the ZnT3 expression in CB?

Author Response

We thank the reviewer for careful and constructive comments.

Comment 1: The authors were sought to investigate the metals distribution in adult mouse brain and spinal cord. Given that the same group has conducted more extensive observation in rat (Ref 18), this mouse study adds little insight in this filed. Here are some comments for improving the manuscript.

Response 1: The study tested whether the findings in the rat were also conserved in mice. The study also excluded the possibility that ZnT3 expression was absent in the spinal cord. Evolutionary conservation of a previously reported difference in one species and excluding one obvious mechanism to explain a difference in metal ion content between two major parts of the central nervous system are two novel conclusions that this paper contributes to the field.

Comment 2: Introduction needs to be reorganized. Current version did not deliver sufficient rationale for repeating the same research from rat to mice.

Response 2: Thank you for this comment, which was also a concern of Reviewer 1. We have now expanded the rationale for our study in the Introduction as follows:

"One of the limitations of our prior work was that it was performed exclusively in only one laboratory animal species, the rat. We therefore readdressed these findings in mice to see if the relative changes in zinc and calcium content between the two parts of the central nervous tissue are evolutionarily conserved between the two rodent species. We report ICP-OES analysis of brain and spinal cord in brains and spinal cords in both sexes of mice and compare the new data to our prior findings in the rat. To address whether lower zinc in the spinal cord was indeed due to the absence of VGLUT1-ZnT3 synaptic vesicles in this tissue, we used a newly developed ZnT3-HA knock-in mouse that attaches a hemagglutinin epitope tag at the C terminus of the endogenous ZnT3 gene to assess the transporter’s abundance in spinal cord sections [56]. Our results, however, suggest that ZnT3 is also expressed in the spinal cord and we therefore discuss the alternative possibility that the lower zinc content per g dry weight in spinal cord versus brain reflects the higher proportion of axonal tracks to cell bodies in the former tissue."

Comment 3: Provide ref of “In cryosections from the brains of ZnT3-HA mice, we found that the ZnT3-HA expression pattern was consistent with previously established data on ZnT3 expression and vesicular zinc concentration and distribution, with high concentrations of ZnT3-HA in the hippocampus, amygdala, and cortex (Figure 3a, left).”

Response 3: We agree with the reviewer for not appropriately placing the references for this claim regarding the distribution of ZnT3 and vesicular zinc. The appropriate references have been added to this claim (Results section, line 226):

[38] Brown, C.E.; Dyck, R.H. Modulation of synaptic zinc in barrel cortex by whisker stimulation. Neuroscience 2005, 134, 355–359.

[57] Palmiter, R.D.; Cole, T.B.; Quaife, C.J.; Findley, S.D. ZnT-3, a putative transporter of zinc into synaptic vesicles. Proc Natl Acad Sci USA 1996,93, 14934–14939.

[58] Brown, C.E.; Dyck, R.H. Distribution of zincergic neurons in the mouse forebrain. J Comp Neurol 2004, 479, 156–167.

Comment 4: The authors claim that: “The working hypothesis that the spinal cord arachnoid accumulates calcium aggregates in …is supported by the ICP-OES analysis presented in this study.” However, I didn’t find this evidence in the present study. The arachnoid was not isolated.

Response 4: Thank you, we agree with your comment. We have now changed the opening statement in the introduction as follows:

“ICP-OES analysis of mouse spinal cords showed high variability in calcium accumulation in this tissue and on average seven-fold more calcium per g dry mass compared to brain. Our interpretation of this finding and working hypothesis is that the spinal cord arachnoid accumulates calcium in form of aggregates in otherwise healthy individual mice, just as we previously demonstrated by X-ray fluorescence imaging in the rat [18].”

Comment 5: Fig 2. Medulla or spinal cord? This figure looks confused in the current version, needs to be represented.

Response 5: Thank you for noticing, we have corrected medulla to spinal cord.

Comment 6: Fig 3. The cerebellum is a major repository of metals, how about the ZnT3 expression in CB?

Response 6: We agree with the reviewer that the cerebellum is a major repository of metals. While we have not analyzed the expression of ZnT3-HA in the cerebellum of our mouseline, evidence from other groups suggests that ZnT3 is only expressed in the cerebellum transiently and at low levels during early development (Valente and Auladell, Molecular and Cellular Neuroscience, 2002). As such, it is unlikely that we would observe ZnT3-HA expression in the cerebellum.

Round 2

Reviewer 2 Report

Comments and Suggestions for Authors

In the revised manuscript, the authors made some improvements to the description. Still, the following points require further clarification:

1.The study's knowledge gap or question would be 1) if the calcium and zinc content of the brain and spinal cord is species-specific or evolutionarily conserved, and 2) whether the distinct concentration of zinc in the brain and spinal cord results from the different expression pattern of ZnT3, the primary transporter vesicle in the CNS. At least in the abstract, should focus on the raising, answering and significance of these two questions.

2. ZnT3-HA reporter line mice only display ZnT3 positive vesicles; not all ZnT3 vesicles overlap with Vglut1+ vesicles.

Author Response

Comments 1:  In the revised manuscript, the authors made some improvements to the description. Still, the following points require further clarification:

1.The study's knowledge gap or question would be 1) if the calcium and zinc content of the brain and spinal cord is species-specific or evolutionarily conserved, and 2) whether the distinct concentration of zinc in the brain and spinal cord results from the different expression pattern of ZnT3, the primary transporter vesicle in the CNS. At least in the abstract, should focus on the raising, answering and significance of these two questions.

Response 1: Thank you. We have changed the abstract according to your suggestion. We are grateful as we agree that the scope of the manuscript is conveyed more clearly following the more explicit wording.

Comments 2: 2. ZnT3-HA reporter line mice only display ZnT3 positive vesicles; not all ZnT3 vesicles overlap with Vglut1+ vesicles.

Response 2: We agree. The sentence starting at line 79 has been modify in the introduction, to include this explicit information. It now reads: "Our finding coincided with another unbiased observation based on proteomic and imaging analysis of single synaptic vesicles isolated from the rat brain concluding that the most abundant (34%) synaptic vesicle type co-expressed vesicular glutamate transporter 1 (VGLUT1) and ZnT3, whereas 25% were of the simple glutamatergic (VGLUT1-positive, ZnT3-negative) type."